# Evaluation of Load Distribution in a Mandibular Model with Four Implants Depending on the Number of Prosthetic Screws Used for OT-Bridge System: A Finite Element Analysis (FEA)

**DOI:** 10.3390/ma15227963

**Published:** 2022-11-10

**Authors:** Francesco Grande, Mario Cesare Pozzan, Raul Marconato, Francesco Mollica, Santo Catapano

**Affiliations:** 1Department of Prosthodontics, University of Ferrara, Via Luigi Borsari 46, 44121 Ferrara, Italy; 2Department of Mechanical and Aerospace Engineering, Politecnico di Torino, Via Nizza, 230, 10126 Torino, Italy; 3Department of Engineering, University of Ferrara, 44121 Ferrara, Italy

**Keywords:** Finite Element Analysis, OT-Bridge, full-arch implant rehabilitation, metal framework materials

## Abstract

In full-arch implant rehabilitations, when the anterior screw abutment channel compromises the aesthetic of the patient, the OT-Bridge system used with its Seeger rings may provide the necessary retention of the prosthesis. However, no studies have evaluated the forces generated at the Seeger level during loading. This Finite Element Analysis aims to investigate the mechanical behavior of Seeger rings in a mandibular model with four implants and an OT-Bridge system, used without one or two anterior prosthetic screws. A 400 N unilateral load was virtually applied on a 7 mm distal cantilever. Two different variables were considered: the constraint conditions using two or three screws instead of four and the three different framework materials (fiberglass reinforced resin, cobalt-chrome, TiAl6V4). The FEA analysis exhibited tensile and compressive forces on the Seeger closest to the loading point. With the resin framework, a tension force on abutment 3.3 generates a displacement from 5 to 10 times greater than that respectively expressed in metal framework materials. In a full-arch rehabilitation with four implants, the case with three prosthetic screws seems to be a safer and more predictable configuration instead of two. Considering the stress value exhibited and the mechanical properties of the Seeger, the presence of only two prosthetic screws could lead to permanent deformation of the Seeger in the screwless abutment closest to the loading point.

## 1. Introduction

Nowadays, completely edentulous jaws can be restored by using removable and fixed prosthetic solutions [1,2,3,4]. The “All-on-Four” protocol is one of the possible treatment modalities for the implant-supported prosthesis [5]. It consists of the insertion of four implants in an interforaminal position, two perpendicular to the occlusal plane and two distally tilted with 30° of angulation [6,7,8].

The connection between implants and prosthesis may foresee the use of different types of abutments [9,10,11,12]. One of the most used anchoring systems is the multi-unit abutment (MUA), which consists of straight or angulated components of different heights that move the implant internal connection to a conical external connection. In this way, a passive prosthetic fit is allowed even in cases of implant disparallelism. Furthermore, the occlusal stress is moved from the implant screw to the multi-unit abutment screw that is smaller than the first one and may be the weak point in case of prosthetic complications [3,10]. Recently a new fixed solution called the OT-Bridge system (Rhein83, Bologna, Italy) was introduced. This system is composed of a low-profile attachment for overdenture, the OT-Equator [13], a sub-equatorial component represented by an interchangeable peek ring called Seeger and a cylindrical titanium “extragrade” abutment which, at its retentive extremity, is provided a cavity designed for the insertion of the Seeger. In this way, the Seeger ring provides an interlocking retention system that guarantees the stable housing of the prosthesis on the OT-Equator attachment.

An important clinical issue, arising with “screw-retained” rehabilitations, especially in the case of severely resorbed jaws, is the presence of a buccal screw access channel that constitutes an anesthetic problem in the anterior area of the mouth. This issue could be managed with a “cemented-retained” solution, although the advantages of the screw-retained prosthesis are lost. Another way to solve this problem could be using angulated abutment but only for less than a 30° disparallelism [14].

Retaining the benefits of a screw-in prosthesis, the OT-Bridge system may represent a solution in the absence of one or two anterior prosthetic screws when a full-arch implant rehabilitation is used. The stability of the OT-Bridge system is provided by both the posterior screws and the interlocking system between the acetal Seeger, inserted in the extragrade abutment, and the subequatorial region of the OT-Equator attachments (Figure 1).

In this way, the OT-Bridge system has been tested and studied [10,15,16,17]. As some in vitro studies demonstrated, this retention system could be used in an All-on-Four rehabilitation also in the absence of one screw among the four provided [16,17]. However, it is not clear if different framework materials play a role in these possible configurations and forces at Seeger level have to be investigated. Knowing the forces expressed at Seeger ring level, in the OT-Bridge system it is important to understand the possibility of this type of rehabilitation on implants. The Seeger ring is necessary to obtain the snap-on retention, independently from the screw insertion. In this way, in an All-on-Four OT-Bridge system rehabilitation, it is interesting to evaluate the forces expressed at this level when one or two anterior screws are not inserted for esthetic issues.

Finite Element Analysis (FEA), as a digital engineering simulation, has been largely used in implant and prosthetic dentistry to study the mechanical behavior of the different parts [18,19]. The most notable advantage of FEA analysis in the biomedical field is the possibility of testing the performance of different devices with different conditions and the distribution and magnitude of stress in the jawbone, implants and prosthetic components [15,20,21]. In the literature, the Finite Element Analysis which investigated the restoration of totally edentulous patients took into consideration different variables such as the framework materials, the number and position of implants and the abutment types [22,23,24]. 

In this study, regarding the use of the OT-Bridge system for full-arch implant rehabilitations where possible esthetic issues could arise from the buccal screw channel made in the prosthesis; we tried to assess some data to understand the forces expressed at the Seeger level. As Seeger ring in the OT-Bridge system has the function of retaining the prosthesis when it is applied alone, without the screw, it is important to know the type and entity of forces expressed at this level during chewing.

This FEA aims to study the amount of forces generated at the Seeger rings applied in an All-on-Four mandible rehabilitation. Two constraint conditions were established using three different framework materials—resin, titanium and Co-Cr—applying a loading on a distal cantilever. The absences of one (constraint condition 1—CC1) or two anterior screws (constraint condition 2—CC2) were studied to understand their effect on the stability of this simulated All-on-Four rehabilitation.

## 2. Materials and Methods

An epoxy resin model of the mandible already described in a previous article [10] was initially scanned with a high precision lab scanner (Optical RevEng, Open technologies S.R.L.; Brescia, Italy) to obtain an STL file of the model. The mandibular geometry was simplified as a rectangular circular object by progressively reducing the number of meshes using a 3D mesh processing open-source software program (MeshLab, ISTI; Pisa, Italy).

According to the “All-on-Four” protocol [6], four implants were virtually inserted into the mandible model with prosthetic emergences in the following teeth positions, according to FDI tooth numbering systems: 35, 33, 43, 45 (left lower canine, 2nd premolar, and right lower canine and 2nd premolar). The anterior implants (NobelParallel, Narrow Platform, 3.75 × 10 mm) were placed perpendicularly to the occlusal plane while the posterior implants (NobelParallel, Narrow Platform, 4.30 × 10 mm) were distally tilted by 30° (Figure 2).

Using an appropriate software program (SolidWorks 2018, Dassault Systèmes SolidWorks Corporation; Vélizy-Villacoublay, France), the implant fixtures, the OT-Bridge abutments and the prosthetic framework were designed. The framework was modeled and drawn with a constant thickness of 4.8 × 5.5 mm and connected to the extragrade abutments (Figure 3).

Then, two different constraint conditions were applied for evaluating the mechanical behaviour of the system:Constraint condition 1 (CC1): 3 screws on 4 implants: 3 joints have been inserted: 2 at the level of the mandibular body to block the mandible in space and 1 at the level of the screwless abutment (4.3).Constraint condition 2 (CC2): 2 distal screws present on 4 implants: 4 joints have been inserted: 2 at the level of the mandibular body to block the mandible in space and 2 at the level of the 2 screwless abutments (3.3 and 4.3).

Three different materials were used for the prosthetic framework: Glass fiber reinforced resin (Trilor Arch, Bioloren S.R.L.; Saronno, Italy);Titanium alloy (Ti6Al4V, Arcam AB; Mölndal, Sweden);Cobalt-chromium (Magnum Lucens, Mesa Italia S.R.L.; Travagliato, Italy) (Table 1).

Combining the three different framework materials with each configuration, a total of six different study models were obtained:CC1 (3 inserted screws) with resin frameworkCC2 (2 inserted screws) with resin frameworkCC1 with titanium frameworkCC2 with titanium frameworkCC1 with Cr-Co frameworkCC2 with Cr-Co framework

A vertical load of 400 N on a 7 mm distal cantilever was applied distally to the implant fixture in 3.5 position. Then, for each framework, the forces generated at the screwless abutments were measured and compared under the two different constraint conditions using ANSYS software (https://www.ansys.com/it-it accessed on 20 April 2022).

In order to study the displacement of the system under loading, especially at Seeger level, constraints have been inserted at the joint’s positions. In this way, the movements exhibited on the XY plane and on the Z-axis were interpreted as the expression of the compressive and tension forces.

## 3. Results

Data obtained are shown in the following tables.

With the resin framework (Table 1), significantly higher forces were generated at 4.3 position in CC1 than in CC2. Forces generated on the Z-axis were always higher than those generated on the other axis in CC1.

In CC2, opposite forces of significant intensity along the Z-axis were generated based on the abutment; traction forces were expressed at 3.3 while compression was at 4.3 position. Forces generated at 3.3 abutment on the XY plane were three times higher (331 N) than those generated in CC1 (99 N) and compared to those at 4.3.

With the titanium framework (Table 2), switching from CC1 to CC2 reduced the force expressed in position 4.3 by about 20 N but a very important force was generated at 3.3. In this position, the force generated was approximately three times greater than that generated in CC1. In CC2, opposite forces of significant intensity along the Z-axis, traction at 3.3 position and compression at 4.3 were observed. On the XY plane in 3.3 position, significantly increased forces were generated, compared to 4.3 position in CC2.

In the Cr-Co framework (Table 3) the resulting force at 4.3 was approximately the same between the two constraint conditions but a very significant resulting force is generated in position 3.3. For this abutment, the 280 N force was 2.5 times greater than the one generated at 4.3 in CC2 (112 N). During the CC2 opposite forces of significant intensity along the Z-axis, traction at 3.3 and compression at 4.3 position were generated. 

For the 3.3 abutment on the XY plane, significantly increased forces were generated during CC2 compared to the abutment in position 4.3 in CC1.

Between the three types of frameworks in CC1, the highest forces were expressed by the Cr-Co (112.37 N) and the titanium framework (109.59 N) at 4.3 abutment. On the other hand, the resin framework generated the highest forces in CC2 at the abutment 3.3 (330.92 N).

## 4. Discussion

In the literature, aesthetic and mechanical problems for fixed implant prostheses have been described [25,26,27,28]. The OT-Bridge system, which provides a retentive propriety in a screw-retained abutment, could represent a good solution to manage the aesthetic problem of vestibular screw canal access by removing the screws [13,16,29,30]. However, it is not clear whether the Seeger ring is able to retain the prosthesis even in absence of one or two anterior screws in an All-on-Four rehabilitation. In this sense, the study of the reaction forces expressed at the Seeger level after the loading application on a distal cantilever could be interesting from a clinical point of view.

This Finite Element Analysis aims to study the entity and direction of forces generated at the Seeger level in a full-arch implant mandibular rehabilitation using the OT-Bridge system in the absence of one (CC1) or two (CC2) anterior prosthetic screws and with three different framework materials.

From the data obtained, changing the framework material and removing one or two screws resulted in significantly different reaction force values. The load, in absence of the two anterior screws, generates tensile and compressive forces on the Seeger at the mesial abutments and a significant increase in the forces on the XY plane closest to the loading point. This could be explained by the type of loading that resulted in a bending moment at the farthest abutments and a compression at the closest abutment from the loading area.

In the resin framework, a horizontal displacement is generated at 3.3 abutment; the movement created is about five times greater than that in the titanium one and about 10 times greater than that in the Co-Cr framework. This could be due to the different rigidity of the material which, as in other studies, demonstrated an increased stability in cases of higher rigidity [31,32]. In addition, in CC2 with the resin framework, the resulting force expressed at 3.3 abutment is more than six times higher than at 4.3, explainable by the distance from the loading point. When comparing 4.3 abutment in CC1 and CC2, the resulting force decreased by 45 N (from 98.56 N to 53.21 N), because of the preservation caused by the Seeger closest to the loading point that absorbed the greatest stress.

A similar pattern of force distribution was also observed in the titanium and Co-Cr frameworks; however, the entity of the forces changed according to the degree of the material rigidity. This is inversely related to the force absorption of the material, which released those forces at the Seeger level of the prosthetic framework. 

In the titanium framework, for CC2, the resulting force at 4.3 is 20 N less than in CC1, but for the Cr-Co framework they are quite equal (112 N in CC1 and 119 N in CC2). This indicates that the Seeger ring closest to the loading point absorbed more stress than a screw, reducing at the same time the entity of the force expressed at the other Seeger ring.

In this view, considering direction, modulus and intensity of the reaction forces and the ratio of these forces to Young’s modulus, which is proportional to displacement, the Seeger ring of the screwless abutment closer to the loading area (abutment in position 3.3) may be deformed. The deformation is quite reliable because of the intensity of the forces generated on the XY plane and on the Z-axis, related to the material of the Seeger that is an acetalic resin.

In constraint condition 1 (CC1), the resulting forces expressed at the abutment in position 4.3 are higher for the Cr-Co framework than the other two materials. Although during CC2 the resin framework and the titanium framework expressed a reduction of 45 and 20 N, respectively, but for the Cr-Co framework a little increase was observed. This could be explained by the absence of elasticity in the Co-Cr material, which leads to poor absorption of the resulting forces by the framework.

In constraint condition 2 (CC2), the highest force values for the abutment in position 3.3 have been observed in the resin framework (330 N) and the lowest in the Co-Cr framework (280 N); this is indicative of the major stability expressed by the Co-Cr framework because of its high rigidity.

Finally, the presence of three instead of four prosthetic screws seems to be a safe and predictable configuration, regardless of the material used for the framework, as the forces expressed could be counteracted by the Seeger ring, and framework disinsertion is quite improbable. As the compression force generated along the Z-axis was seen for all three of the materials, the highest intensity of forces was found with the resin framework due to lower mechanical properties. In this type of framework, significantly larger displacements were generated, especially, on the XY plane, probably leading to permanent deformation of the acetalic Seeger instead of the titanium or the Co-Cr one.

Then, it is important to consider the biomechanical response of the Seeger ring in light of the framework material used. In fact, the type of framework material, depending on the different Young’s modulus, influences the tension and deformation forces expressed at the Seeger level. This could lead to Seeger ring deformation, fracture and possibly prosthetic disinsertion, in cases of screwless abutment.

The limits of this Finite Element Analysis mainly involved the virtual nature of the study. The absence of an in vivo environment involving the presence of saliva or other oral fluids makes necessary a clinical implementation of this study. The results were also conditioned by the parameters used such as prosthetic framework materials and loading conditions. Variations of those factors may lead to different forces in terms of direction, intensity and displacements and also the biomechanical behavior of the prosthetic components may be different. In addition, the mandibular constraints, which were inserted at the level of the mandibular body and not in the mandibular condyles, the design of the jawbone and of the prosthetic framework and the analysis of only one loading condition led to caution in the interpretation of these initial results. The screwed abutment has been considered fixed without the effect of the constraint forces which could dissipate in a different way the forces on the other abutments. This is the reason why the condition with all four of the screws inserted was not studied.

In addition, the effects of the preload torque of the screws were not considered and the materials from a mechanical point of view were supposed to be linear, elastic and isotropic.

## 5. Conclusions

In conclusion, the OT-Bridge system with an All-on-Four rehabilitation seems to be stable and safe in absence of one anterior screw (CC1). The stress was higher for the Co-Cr framework; however, its results are more balanced compared with the other two materials, especially in CC2. This could prevent the deformation of the Seeger and of the framework, favoring the stability of the prosthetic rehabilitation. Among the results of this study, prosthetic rehabilitations with an aesthetic screw channel could be managed with the absence of one prosthetic screw using the OT-Bridge system. These results are also conditioned by constraints and other simplifying assumptions; the variation of these parameters certainly leads to a variation of forces in terms of direction, intensity and displacements. In addition, the screwless abutment was assumed as a single block with the Seeger, but the acetalic Seeger propriety, which is elastic, has not been considered.

Therefore, based on these assumptions, further in vitro and in vivo studies are needed to verify the results obtained and to test the effective biomechanical behaviour and resistance of the Seeger ring. Interesting opportunities in implant patients can be reached with the All-on-Four rehabilitation using the OT-Bridge system.

## Figures and Tables

**Figure 1 materials-15-07963-f001:**
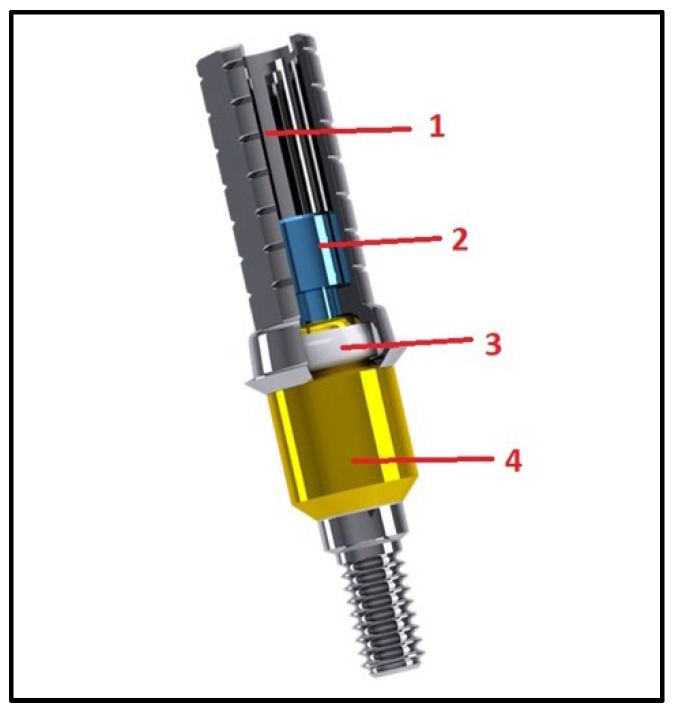
The OT-Bridge system (Rhein83, Bologna, Italy) composed with the extragrade (1), the prosthetic screw (2), Seeger (3), OT-Equator (4).

**Figure 2 materials-15-07963-f002:**
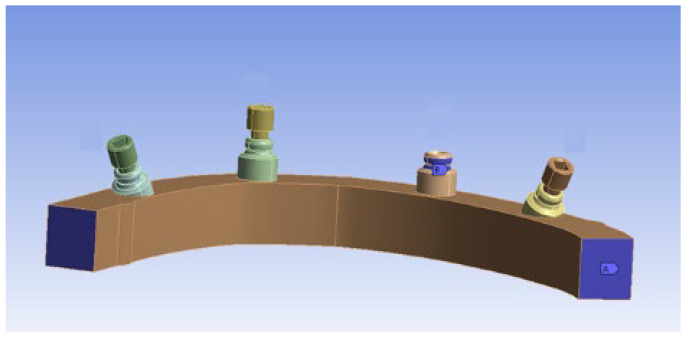
The simplified mandible model with implants and OT-Equator inserted and only 3 prosthetic screws.

**Figure 3 materials-15-07963-f003:**
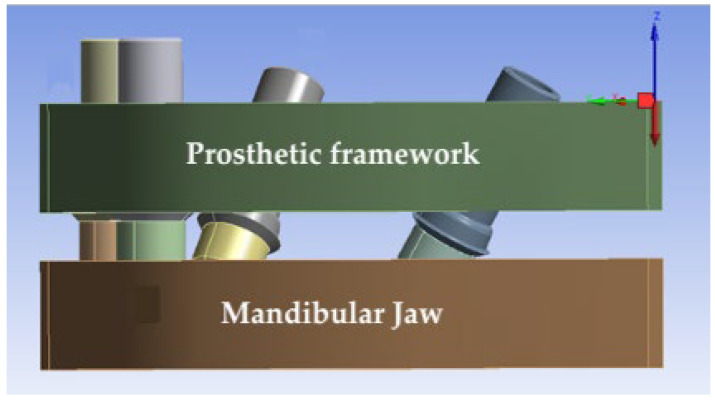
The mandible model and the prosthetic framework connecting the extragrade abutments to the OT-Equator attachments. The load is applied on the framework cantilever and is indicated in red.

**Table 1 materials-15-07963-t001:** Results of FEA evaluation for the resin framework in CC1 and CC2 R = resulting (reaction) force; X,Y,Z = (reaction) components of the resulting force; unit of measurement = reaction force (N).

Resin Framework	CC1	CC2	
	Abutment 4.3	Abutment 3.3	Abutment 4.3
X-axis	38.60 N	228.28 N	−8.47 N
Y-axis	63.29 N	221.20 N	−16.40 N
Z-axis	92.00 N	−49.90 N	−64.95 N
Resulting	98.56 N	330.92 N	53.21 N

**Table 2 materials-15-07963-t002:** Results of FEA evaluation for the TiAl6V4 framework in CC1 and CC2 R = resulting (reaction) force; X,Y,Z = (reaction) components of the resulting force; unit of measurement = reaction force (N).

TiAl6V4 Framework	CC1	CC2	
	Abutment 4.3	Abutment 3.3	Abutment 4.3
X-axis	63.53 N	165.49 N	17.77 N
Y-axis	76.40 N	196.39 N	−12.09 N
Z-axis	−46.84 N	139.60 N	−86.85 N
Resulting	109.59 N	292.31 N	89.47 N

**Table 3 materials-15-07963-t003:** Results of FEA evaluation for the Co-Cr framework in CC1 and CC2 R = resulting (reaction) force; X,Y,Z = (reaction) components of the resulting force; unit of measurement = reaction force (N).

Cr-Co Framework	CC1	CC2	
	Abutment 4.3	Abutment 3.3	Abutment 4.3
X-axis	68.52	132.12	27.92
Y-axis	80.96	175.11	−0.14
Z-axis	−37.11	174.61	−115.90
Resulting	112.37	280.37	119.22

## Data Availability

Not applicable.

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
