# Peer review of "Evaluation of Load Distribution in a Mandibular Model with Four Implants Depending on the Number of Prosthetic Screws Used for OT-Bridge System: A Finite Element Analysis (FEA)"

_materials, 2022, doi:10.3390/ma15227963_

Round 1

Reviewer 1 Report (Previous Reviewer 3)

In this study, the researchers study the load distribution in an All-on-4 model depending on the number of prosthetic screws used for OT-Bridge system using the Finite Element Analysis.

It will be better to understand if the authors can add OT-Bridge system (Rhein83, Bologna, Italy) Figure.

“Finite Element Analysis (FEA), as a digital engineering simulation, has been largely used in implant and prosthetic dentistry to study the mechanical behavior of the different parts.”

Please add References https://doi.org/10.3390/app11031220

doi: 10.7860/JCDR/2013/7001.3775

Method

The methods are not clear and bit confusing. So, the method section needs editing and simplification.

Conclusion:

Add in which case among the three different framework materials of resin, titanium and Co-Cr, stress is less?

Author Response

Dear reviewer,

thanks for your revision. Here you can read a point-to-point answer to your suggestions and comments.

In this study, the researchers study the load distribution in an All-on-4 model depending on the number of prosthetic screws used for OT-Bridge system using the Finite Element Analysis.

It will be better to understand if the authors can add OT-Bridge system (Rhein83, Bologna, Italy) Figure. We added a figure of the Ot-Bridge system in the introduction

“Finite Element Analysis (FEA), as a digital engineering simulation, has been largely used in implant and prosthetic dentistry to study the mechanical behavior of the different parts.”

Please add References https://doi.org/10.3390/app11031220

doi: 10.7860/JCDR/2013/7001.3775
We added those references at that point you want

Method

The methods are not clear and bit confusing. So, the method section needs editing and simplification.

We edited and simplified the materials and methods section especially in the last part making it more understandable

Conclusion:

Add in which case among the three different framework materials of resin, titanium and Co-Cr, stress is less?

In the conclusion we specified "In conclusion, the OT-Bridge system with an All-On-Four rehabilitation seems to be stable and safe in absence of one anterior screw (CC1). The stress was higher for the Co-Cr framework, however it results more balanced compared with the other two materials, especially in CC2. This could prevent the deformation of the Seeger and of the framework, favoring the stability of the prosthetic rehabilitation"

Reviewer 2 Report (Previous Reviewer 2)

It was evaluated the article titled “Evaluation of load distribution in an All-on-4 model depending on the number of prosthetic screws used for OT-Bridge system: a FEM analysis”, comparing 3 different materials. This study was resubmitted (previous submission number was materials-1888594.

This is an interesting study. Therefore, there are many concerns.

- All-on-4 is a trademark. Have the authors authorization to use here? Otherwise, I suggest to change the terminology.

- tables: substitute comma by point in the numbers.

- Abstract, Intro, and Discussion: improved and it is satisfactory

- M&M: although there were used 3 different materials, there is a failure in this study. Where is the control group with all screws and where is the lack of screw in the distal implants? Why was not tested the lack of 1 or 2 screws in other positions? I suggested to consider 5 groups in your study (complement the study which is a lab/silica study).

- Conclusion: I consider it incomplete due to lack of groups.

Author Response

It was evaluated the article titled “Evaluation of load distribution in an All-on-4 model depending on the number of prosthetic screws used for OT-Bridge system: a FEM analysis”, comparing 3 different materials. This study was resubmitted (previous submission number was materials-1888594.

This is an interesting study. Therefore, there are many concerns.

- All-on-4 is a trademark. Have the authors authorization to use here? Otherwise, I suggest to change the terminology.
Dear reviewer, we substituted this terminology in the title. However, our model was produced according to the all-on-four protocol and we cited the appropriate articles all over in the text.

- tables: substitute comma by point in the numbers.
we substituted comma by points

- Abstract, Intro, and Discussion: improved and it is satisfactory

- M&M: although there were used 3 different materials, there is a failure in this study. Where is the control group with all screws and where is the lack of screw in the distal implants? Why was not tested the lack of 1 or 2 screws in other positions? I suggested to consider 5 groups in your study (complement the study which is a lab/silica study).
Dear reviewer, we specified in the introduction that "An important clinical issue, arising with “screw-retained” rehabilitations, especially in the case of severely resorbed jaws, is the presence of a buccal screw access channel that constitutes an anesthetic problem in the anterior area of the mouth.
Retaining the benefits of a screw-in prosthesis, the OT-Bridge system may represent a solution in the absence of one or two anterior prosthetic screws". 
This is the only case where the OT-Bridge system can be used without screw. Then, there is no clinical sense in using the OT-Bridge system without screws in the posterior area, because the posterior teeth are not involved in the aesthetics. Also the manufacturer does not consider the use of OT-Bridge system without posterior screws because it is 
senseless from a clinical point of view.

For the group with all the four screws, as 
we reported in the limits of this study:“The screwed abutment has been considered fixed without the effect of the constraint forces which could dissipate in a different way the forces on the other abutments. This is the reason why the condition with all the 4 screws inserted wasn’t studied.”

- Conclusion: I consider it incomplete due to lack of groups.
Dear reviewer, we changed the conclusion section according also to another reviewer, specifying in which case the forces are more or less expressed.

Reviewer 3 Report (Previous Reviewer 1)

The authors tried to address the comments, and the manuscript improved compared to the earlier version. The authors didn't answer or address several comments. Kindly find below all the comments pay attention and address them

Abstract:

1.       Kindly mention the significant values

2.       Kindly provide the MesH keywords

Introduction:

1.       It seems that the research question it attempted to answer is still not clearly defined or adequately justified

Methods:

1.       3.5, 3.3, 4.3, 4.5. : response: We clarified it, writing that indicates the teeth positions: what do the authors mean by this? Teeth position?

2.       Kindly illustrate the methods to make it reader easy: this means providing a conceptual framework or figure

Results:

1.       The results were reported mainly based on virtual software, depending on the materials used and various points in the mandible model. Hence the results presented were from virtual software, a single force. Is the result enough to justify the research questions or the hypothesis?

Discussion:

1.       However, it is not clear whether the Seeger ring can retain the prosthesis even in the absence of one or two anterior screws in an all-on-four rehabilitation: this statement of the authors is in contrast to the current study

2.       If points 1,2 and 3 are the limitations of the study, kindly justify how and what this study will add to the clinical practice and the literature

3.       The research question is not clearly described as well is the new information supplied on implant practice clinically: This has not been answered

4.       The outcomes presented on assumptions: provide a robust rationale: This has not been answered

Author Response

The authors tried to address the comments, and the manuscript improved compared to the earlier version. The authors didn't answer or address several comments. Kindly find below all the comments pay attention and address them

Abstract:

  1. Kindly mention the significant values
    Dear reviewer, in the abstract we write that " In a full-arch rehabilitation with four implants, the case with three prosthetic screws seems to be a safer and more predictable configuration instead of two. Considering the stress value exhibited and the mechanical properties of the Seeger, the presence of only two prosthetic screws could lead to permanent deformation of the seeger in the screw-less abutment closest to the loading point".
    In this way, from a clinical view point, we expressed the scientific value of our study

  2. Kindly provide the MesH keywords
    Dear reviewer, we provided the MesH Keywords in the abstract

Introduction:

  1. It seems that the research question it attempted to answer is still not clearly defined or adequately justified
    From the previous version of the manuscript we largely modified the introduction part. We try to better express the research question by explaining the importance of knowing the entity and type of forces expressed at the seeger rings in an all-on-four rehabilitation with only two or three prosthetic screws inserted. 

Methods:

  1. 3.5, 3.3, 4.3, 4.5. : response: We clarified it, writing that indicates the teeth positions: what do the authors mean by this? Teeth position?

    Tooth numbering system is used by dentists for uniquely identifying and referring to a specific tooth. Over the years, over 20 different teeth numbering systems have been developed. Today, we use the following FDI tooth numbering systems for the numbering of the teeth. FDI World Dental Federation Two-Digit Notation (international). This system developed by the Fédération Dentaire Internationale (FDI), World Dental Federation notation is also known as ISO-3950 notation. The human teeth are symmetrically arranged in the mouth. Each quadrant of the mouth has 8 different teeth that are mirrored horizontally and vertically to the other quadrants. In the FDI (Fédération Dentaire Internationale) World Dental Federation notation each one of these 8 teeth is assigned a number from 1 to 8, starting from the center front tooth (central incisor) and moving backwards up to the third molar (number 8). Each quadrant is also assigned a number, from 1 to 4 for the adult (permanent) teeth or 5 to 8 for the baby (primary or deciduous) teeth. The combination of these two numbers (Quadrant code number & Tooth code number) specifies how are teeth numbered. This tooth numbering system is called, the Two-Digit World Dental Federation Notation or FDI notation system.
    We inserted in materials and methods that we used the FDI method

  2. Kindly illustrate the methods to make it reader easy: this means providing a conceptual framework or figure
    Dear reviewer, we provided the figure 3 with the explanation of what is the prosthetic framework and of the mandibular jaw. It is possible to see that the Extragrade abutment (peculiar of the OT-Bridge system) connects the OT-Equator attachments with the prosthetic framework. We also specified that the load is applied on the distal cantilever and it is indicated in red

Results:

  1. The results were reported mainly based on virtual software, depending on the materials used and various points in the mandible model. Hence the results presented were from virtual software, a single force. Is the result enough to justify the research questions or the hypothesis?
    Thanks for this comment. We analyzed the forces expressed at Seeger ring level, that, in the OT-Bridge system is necessary to obtain the snap-on retention, independently from the screw insertion. In this way, in an all-on-four OT-Bridge system rehabilitation, it is interesting to evaluate the forces expressed at the Seeger ring level when one or two anterior screws are not inserted for esthetic issues. We specified it in the introduction

Discussion:

  1. However, it is not clear whether the Seeger ring can retain the prosthesis even in the absence of one or two anterior screws in an all-on-four rehabilitation: this statement of the authors is in contrast to the current study

    Dear reviewer, as it is a FEA analysis we have to tone down conclusion about this aspect. However, from the data obtained we can conclude that the all-on-four with only one screw left on the to-bridge is a stable and safe solution, which means that the seeger ring provides the retention of the prosthesis with 3 screw inserted. We write this in the abstract and also in the conclusion

  2. If points 1,2 and 3 are the limitations of the study, kindly justify how and what this study will add to the clinical practice and the literature

    This study shows to the clinicians the constraint forces in the case of an aesthetic screw channel could be managed with the absence of one prosthetic screw using the OT-Bridge system using different frameworks. By the way using a resin framework in this solution could lead to a permanent deformation of the seeger ring. We added this point in the text

  3. The research question is not clearly described as well is the new information supplied on implant practice clinically: This has not been answered

    In the implant and clinical practice an aesthetic screw channel access it’s a important and difficult issue to managed and this study supports that the OT-Bridge solution is safe and predictable when a seeger ring is left alone without one prosthetic screw. 
    In addition, using a Co-Cr framework could prevent the deformation of the Seeger, favoring the stability of the prosthetic rehabilitation. Among the results of this study, than we could consider that the prosthetic rehabilitations with an aesthetic screw channel could be managed with the absence of one prosthetic screw using the OT-Bridge system with a Cr-Co framework. That is what we added in the text

  4. The outcomes presented on assumptions: provide a robust rationale: This has not been answered
    Thanks for this comment. We improved the explanations associated to our outcomes

Round 2

Reviewer 1 Report (Previous Reviewer 3)

Many thanks for the revision and incorporating all suggested changes to the manuscript

Author Response

Dear reviewer,

thanks for your kind and useful revisions. We greatly improved our work also thanks to you. 
Best regards

Reviewer 2 Report (Previous Reviewer 2)

It was evaluated the article titled “Evaluation of load distribution in a mandibular model with four implants depending on the number of prosthetic screws used for OT-Bridge system: a Finite Element Analysis (FEA)”, comparing 3 different materials. This study was resubmitted (previous submission number was materials-1888594) - Version 2.

I appreciated all responses sent. I considered the article enough to be published.

Congratulations.

Author Response

Dear reviewer,

I really appreciated your comments and suggestions for improving our work.
Thank you 
Best regards

Reviewer 3 Report (Previous Reviewer 1)

Dear Authors,

The authors have addressed all the comments and the manuscript is much improved and now seems well presented. I would like to congratulate the authors for their work and wish them all the very best for future endeavors.

Kindly address and change the following comment in the method section:

1. 3.5, 3.3, 4.3, 4.5. : response: We clarified it, writing that indicates the teeth positions: what do the authors mean by this? Teeth position?

It should be written as 33,35,43,45 (left lower canine, 2nd premolar, and right lower canine and 2nd premolar). without the dot 

Author Response

Dear reviewer,

we erased the dot from the teeth numbering and inserted what teeth are indicated with those numbering, according to your proposal.
Thank you very much for your collaborative and helpful revision
Best regards

This manuscript is a resubmission of an earlier submission. The following is a list of the peer review reports and author responses from that submission.

Round 1

Reviewer 1 Report

Dear Authors,

I read the manuscript with great interest. The manuscript under review attempts to evaluate virtually the load distribution in an All-on-4 model depending on the number of prosthetic screws used for the OT-Bridge system using FEM analysis. All the sections of the manuscript need several corrections. Kindly find below the manuscript's detailed comments and suggestions, which will help the authors check and revise the manuscript.

Title: kindly write the complete form: Finite Element Analysis (FEA) or Finite Element Method (FEM) for a better understanding of readers

Abstract:

1.     FEM: Kindly write the complete form where it appears first and followed by the short form throughout the text

2.     Kindly provide structured abstract

3.     Kindly mention the significant values

4.     Kindly provide the MesH keywords

Introduction:

1.     FEM: Kindly write the complete form where it appears first and followed by the short form throughout the text

2.     The introduction lacks some essential information. Improve the rationale of the study

3.     Provide a clear justification of the study, including the selected outcomes

4.     Briefly write about the literature on fixed prosthetics, screw supports and bridges.

5.     It seems that the research question it attempted to answer is still not clearly defined or adequately justified

 Materials and Methods:

1.     An epoxy resin model of the mandible: provide details

2.     3.5, 3.3, 4.3, 4.5. : provide details

3.     The constraining force was set as proportional to a body's movement in space: explain and provide details. References?

4.     Kindly illustrate the methods to make it reader easy

Results:

I found that the results were reported mainly based on virtual software, depending on the materials used and various points in the mandible model. Hence the results presented were from virtual software, a single force. Is the result enough to justify the research questions or the hypothesis?

Discussion:

1.     The discussion must acknowledge the implications of the findings for clinical practice. The implants in dentistry have a more sophisticated rationale, and it's a surgical and prosthetic procedure. Hence outcomes of the research should focus on the clinical implementations: discuss in this regards

2.     The main concern of the study is that it was virtually carried out

3.     Proper research based on elements and materials used expected with in-vivo conditions for implants: kindly discuss and provide your explanation

4.     The research question is not clearly described as well as the new information supplied on implant practice clinically

5.     The outcomes presented on assumptions: provide a robust rationale.

Clinical implementation: kindly provide a strong justification in the subheading

Future recommendations and clinical studies: kindly provide a strong justification in the subheading

Concussion: rewrite the conclusions after addressing all the comments

Author Response

Dear reviewer, thanks for these comments and suggestions. We rewrite the article according to your revision. Please see below a point-to point answer to your comments.

Dear Authors,

I read the manuscript with great interest. The manuscript under review attempts to evaluate virtually the load distribution in an All-on-4 model depending on the number of prosthetic screws used for the OT-Bridge system using FEM analysis. All the sections of the manuscript need several corrections. Kindly find below the manuscript's detailed comments and suggestions, which will help the authors check and revise the manuscript.

Title: kindly write the complete form: Finite Element Analysis (FEA) or Finite Element Method (FEM) for a better understanding of readers
Thanks for this comment. We correct this error specifying the type of work.

Abstract:

  1. FEM: Kindly write the complete form where it appears first and followed by the short form throughout the text
    Thanks for this comment. We write the complete form in the abstract and all-over the text
  2. Kindly provide structured abstract
    Thanks for this comment. According to the journal guidelines, we provide a structured abstract but without subheadings
  3. Kindly mention the significant values
    Thanks for this comment. We mentioned the significant results of the study
  4. Kindly provide the MesH keywords
    Thanks for this comment. We provide the abstract with the MesH keywords

Introduction:

  1. FEM: Kindly write the complete form where it appears first and followed by the short form throughout the text
    Thanks for this comment. We corrected this point
  2. The introduction lacks some essential information. Improve the rationale of the study
    Thanks for this comment. We improved the introduction highlighting the rational of the study
  3. Provide a clear justification of the study, including the selected outcomes
    Thanks for this comment. We highlighted in a more explicit way why we conducted this study, highlighting the connection to the rational of the study
  4. Briefly write about the literature on fixed prosthetics, screw supports and bridges.
    Thanks for this comment. We improved the introduction by talking also about these arguments
  5. It seems that the research question it attempted to answer is still not clearly defined or adequately justified
    Thanks for this comment. We clarified it by highlighting the rational of the study

 Materials and Methods:

  1. An epoxy resin model of the mandible: provide details
    Thanks for this comment. We clarified it
  2. 3.5, 3.3, 4.3, 4.5. : provide details
    Thanks for this comment. We clarified it, writing that indicates the teeth positions
  3. The constraining force was set as proportional to a body's movement in space: explain and provide details. References?
    Thanks for this comment. In order to measure the displacement and the forces expressed in a system, it is necessary to set this condition in the computer software
  4. Kindly illustrate the methods to make it reader easy
    Thanks for this comment. We illustrated step-by-step the methods used for carrying out the study

Results:

I found that the results were reported mainly based on virtual software, depending on the materials used and various points in the mandible model. Hence the results presented were from virtual software, a single force. Is the result enough to justify the research questions or the hypothesis?
Thanks for this comment. We analyzed the forces expressed at Seeger ring level, that, in the OT-Bridge system is necessary to obtain the snap-on retention, independently from the screw insertion. In this way, in an all-on-four OT-Bridge system rehabilitation, it is interesting to evaluate the forces expressed at the Seeger ring level when one or two anterior screws are not inserted for esthetic issues. We specified it in the introduction

Discussion:

  1. The discussion must acknowledge the implications of the findings for clinical practice. The implants in dentistry have a more sophisticated rationale, and it's a surgical and prosthetic procedure. Hence outcomes of the research should focus on the clinical implementations: discuss in this regards
    Thanks for this comment. We discuss the outcomes of the study in a more clinical view, linking every result to a possible clinical outcome
  2. The main concern of the study is that it was virtually carried out
    Thanks for this comment. We improved the discussion by adding the limits of the study and highlighting this main concern
  3. Proper research based on elements and materials used expected with in-vivo conditions for implants: kindly discuss and provide your explanation
    Thanks for this comment. We highlighted those points in the limits of the study
  4. The research question is not clearly described as well as the new information supplied on implant practice clinically
    Thanks for this comment. We improved both the introduction and the discussion to provide a more robust research question in view of the possible clinical advantages
  5. The outcomes presented on assumptions: provide a robust rationale.
    Thanks for this comment. We improved the explanations associated to our outcomes

Clinical implementation: kindly provide a strong justification in the subheading
Thanks for this comment. We improved this part specifying the possible clinical opportunities

Future recommendations and clinical studies: kindly provide a strong justification in the subheading
Thanks for this comment. We supported these future recommendations

Reviewer 2 Report

It was evaluated the article titled “Evaluation of load distribution in an All-on-4 model depending on the number of prosthetic screws used for OT-Bridge system: a FEM analysis”, comparing 3 different materials.   This is an interesting study. Therefore, Thera are many concerns about it.  
  • abstract: the description of materials and methods is incomplete; avoid to use acronyms without to cite before (FEM)
  • Intro: can be improved
  • M&M: although there were used 3 different materials, there is a failure in this study. Where is the control group with all screws? Why was not tested the lack of 1 or 2 screws in other positions? I suggested to consider 5 groups in your study.
  • Discussion: well done
  • Conclusion: I consider it incomplete

Author Response

It was evaluated the article titled “Evaluation of load distribution in an All-on-4 model depending on the number of prosthetic screws used for OT-Bridge system: a FEM analysis”, comparing 3 different materials.   This is an interesting study. Therefore, Thera are many concerns about it.  
Dear reviewer, thanks for these comments. We improved each single section of the article according to your comments and suggestions

  • abstract: the description of materials and methods is incomplete; avoid to use acronyms without to cite before (FEM)
    Thanks for this comment. We correct this error specifying the type of work before the acronyms
  • Intro: can be improved
    Thanks for this comment. We improved the introduction by clinically justifying the reason for this study
  • M&M: although there were used 3 different materials, there is a failure in this study. Where is the control group with all screws? Why was not tested the lack of 1 or 2 screws in other positions? I suggested to consider 5 groups in your study.
    Thanks for this comment. We did not perform the experiment removing the screws in other positions than the anterior ones because from a clinical point of view there is no need to leave the abutment screwless in a posterior position. We improved the introduction explaining in a more evident way that the reason to close the screw access hole in a tooth of the prosthesis is only for esthetic issues.
  • Discussion: well done
  • Conclusion: I consider it incomplete
    Thanks for this comment. We improved the conclusion by adding the clinical rational and possible clinical outcomes of this study

Reviewer 3 Report

The research should be redesigned with expanding variables and evaluation parameters.

Author Response

The research should be redesigned with expanding variables and evaluation parameters

Dear reviewer, thanks for your comment. In this study we focused on the all-on-four rehabilitation with OT-Bridge system and we applied two variables that are the framework material and the constraint condition (by simulating the removal of one or two anterior prosthetic screws in view of a possible esthetic problem). Then, we observed what happens to the seeger ring, that is peculiar of the OT-Bridge system. The comprehension of the forces expressed at this level, with these configurations (absence of 1 or 2 anterior prosthetic screws) are of high importance from a prosthetic point of view. It is important to know the stability of the system and how the seeger ring is stressed when the screw connecting the prosthesis to the implant attachment is absent. Studying what happens at bone and at implants level was not so interesting and was already done by other authors. However, it could be also be performed in another studies